# Reversible First-Order Single Crystal to Single Crystal Thermal Phase Transition in [(CH_3_)_3_CNH_3_]_4_[V_4_O_12_]

**DOI:** 10.3390/ma15165663

**Published:** 2022-08-17

**Authors:** Pablo Vitoria, Ana San José Wéry, Leire San Felices, Laura Bravo-García, Estibaliz Ruiz-Bilbao, José Manuel Laza, José Luis Vilas, Juan M. Gutiérrez-Zorrilla

**Affiliations:** 1Departamento de Química Orgánica e Inorgánica, Facultad de Ciencia y Tecnología, Universidad del País Vasco UPV/EHU, 48080 Bilbao, Spain; 2Facultad de Ciencias y Artes, Universidad Católica de Ávila, c/Canteros s/n, 05005 Ávila, Spain; 3Servicios Generales de Investigación SGIker, Facultad de Ciencia y Tecnología, Universidad del País Vasco UPV/EHU, 48080 Bilbao, Spain; 4Departamento de Química Física, Facultad de Ciencia y Tecnología, Universidad del País Vasco UPV/EHU, 48080 Bilbao, Spain; 5BCMaterials, Parque Tecnológico de Bizkaia, Edificio 500, 48160 Derio, Spain

**Keywords:** polyoxovanates, SCSC transformations, thermal and structural characterization

## Abstract

The well-known compound tetrakis(tert-butylammonium)-cyclo-tetrametavanadate (V), [(CH_3_)_3_CNH_3_]_4_[V_4_O_12_] (1h_RT), which crystallizes in the tetragonal I4/m space group, undergoes an irreversible solid state transformation upon heating, constituting one of the few examples in which the initial and the final stages are structurally characterized by sc-XRD. Now, we observed the ability of the same compound to undergo an additional single-crystal-to-single-crystal (SCSC) transformation upon thermal stimuli, but this time at low temperatures (153 K). Compound 1h_RT contains a discrete unprotonated [V_4_O_12_]^4−^ tetrahedral anion in which V and O bridging atoms are coplanar. In both phases, these tetrameric anions are linked through tert-butylammonium cations in an extensive network of hydrogen bonds, but at low temperatures, this phase loses its characteristic O-V-O coplanarity, with the resulting rearrangement of the crystal packing and hydrogen-bond network which provide its reversibility at low temperatures. Again, the initial and final stages have been characterized structurally by sc-XRD.

## 1. Introduction

Solid-state phase transformations given by an external stimuli such as temperature or pressure [1,2,3,4], in which the integrity of the crystal is retained, have recently acquired remarkable interest as they constitute a promising research field within materials science [5,6,7]. Such transformations, so-called Single-Crystal-to-Single-Crystal (SCSC) transformations, usually involve several modifications within the crystal packing of a structure, leading to switchable materials able to reversibly modify their physical or chemical properties between two or more relatively stable states [8,9]. In this sense, the modification of the inner arrangement of a system implies significant effects on their inherent properties such as color, magnetism, luminescence, or adsorption capacities [10,11,12]. Furthermore, SCSC transformations enables an unequivocal determination of the exact location of atoms and molecules within the crystal packing of the second phase, which allows for observing the changes that undergo that kind of structural transition and, thus, identify how an external stimulus can influence a property of interest to selectively tune it. Due to these characteristics, these materials have attracted great attention in materials exhibiting fine-tunable applicabilities, as the physicochemical properties can be modulated on demand, which makes them applicable in various fields such as electronic devices, sensors, or storage devices [13,14,15].

In this context, polyoxometalates (POMs) constitute ideal building blocks as these inorganic metal-oxo clusters show structural and compositional versatility with a wide range of both physical and chemical properties, with important applications in several fields such as material science (magnetism, luminescence, catalysis…) [16,17,18] and biomedicine [19,20]. In particular, polyoxovanadates (POVs) represent an interesting group due to: (i) the variable coordination geometries of the vanadium centers that include {VO_4_} tetrahedra, {VO_5_} square pyramids, or {VO_6_} octahedra; and (ii) the oxidation state of the vanadium centers that can range from 5+ to 3+ and usually leads to species where they coexist in different ratios as a function of the pH. Despite these features and the fact that SCSC transformations in POM-based systems have long been known, the number of solid-phase transitions reported is lower compared to those for Metal-Organic Frameworks (MOFs) or Porous Coordination Polymers (PCPs) [21,22].

Herein, we report on the thermally triggered structural modification of the already known [(CH_3_)_3_CNH_3_]_4_[V_4_O_12_] (**1h_RT**) at low temperatures. In a previous work [23], we described how **1h_RT** loses its crystallinity at high temperatures due to an irreversible phase transition in a solid state by means of a nucleation and growth mechanism. However, the initial and final stages were able to be structurally characterized by single-crystal X-ray diffraction (sc-XRD), being the phase obtained at high temperatures characterized only after recrystallization in water. In this way, we could determine the rearrangement of the cyclic tetrametavanadate [V_4_O_12_]^4-^ species into a metavanadate form based on [VO_3_]^-^ anions, which is representative of being one of the first solid-state transformations in POM-based systems. In the present work, we also study the thermally-triggered structural trans-formation that **1h_RT** is able to undergo, but this time at low temperatures. Fortunately, the resulting reversible solid-state transitions could be easily followed by sc-XRD techniques. In this case, this transformation is reversible and involves the loss of planarity of the O-V-O bonds, which occurs through a thermal stimulus, after the reorganization of the hydrogen bonds.

## 2. Experimental

### 2.1. Materials and Methods

The compound [(CH_3_)_3_CNH_3_]_4_[V_4_O_12_] was synthesized according to the literature methods and identified by infrared (FT–IR) spectroscopy [23], and this data is as well provided as Appendix A. Fourier Transformed Infrared (FT-IR) spectra were obtained as KBr pellets on a Shimadzu FTIR-8400S spectrometer (Appendix A). Thermal properties of the sample (8.3 mg) were measured by Differential Scanning Calorimetry (DSC 822e from Mettler Toledo) in an aluminum pan under constant nitrogen flow (20 mL·min^−1^). Compound [(CH_3_)_3_CNH_3_]_4_[V_4_O_12_] was subjected to a cooling/heating program: heating from −150 to −90 °C at a rate of 0.2 °C.min^−1^, followed by a cooling scan from −80 to −150 °C at a rate of −0.2 °C.min^−1^ (Appendix A.)

### 2.2. X–Ray Crystallography

Crystallographic data for the low and high temperature phases of the [(CH_3_)_3_CNH_3_]_4_[V_4_O_12_] compound are summarized in Table 1. For single-crystal X-ray diffraction measurements, the crystals were heated at different temperatures; (i) cooling from 173 to 133 K every 10 K, and (ii) heating from 143 to 183 K every 10 K, and then quenched to 100 K to 150 K for the data collection. Intensity data were collected on an Agilent Technologies SuperNova diffractometer, equipped with monochromatic Mo Kα radiation (*λ* = 0.71073 Å) and Eos CCD detector. Data frames were processed (unit cell determination, analytical absorption correction with face indexing, intensity data integration and correction for Lorentz and polarization effects) using the CrysAlis software package [24]. The structures were solved using OLEX2 [25] and refined by full-matrix least-squares with SHELXL–2014/6 [26]. Final geometric calculations were carried out with Mercury [27] and PLATON [28] as integrated in WinGX [29], and their visualization performed with CrystalMaker [30]. The ORTEP representation of **1-HTP** and **1-LTP** were also represented in CrystalMaker, showing a 50% probability thermal ellipsoid. Thermal vibrations were treated anisotropically for all non-hydrogen atoms. Hydrogen atoms of the organic ligands were placed at calculated positions and refined using a riding model with standard SHELXL parameters. These data can be obtained free of charge from The Cambridge Crystallographic Data Centre [31].

The single-crystals were subjected to a cooling-heating procedure before X-ray data collection. In this way, the samples that were collected under the cooling process of the samples were named as: **1c_173, 1c_163; 1c_153; 1c_143; 1c_133**, and the ones collected throughout the heating process were named as: **1h_143, 1h_153, 1h_163, 1h_173, 1h_RT**. Nevertheless, for more clarity, we classify these 10 measurements into two groups: the crystallographic phase obtained at high temperatures (**1-HTP**), which exhibit the space group *I4/m* and comprises the identification codes **1c_173, 1c_163; 1c_153, 1h_163, 1h_173, 1h_RT**, and the crystallographic phase obtained at low temperatures (**1-LTP**), which exhibit the space group *I4/a* and comprises the identification codes **1c_143; 1c_133, 1h_143, 1h_153**.

## 3. Results and Discussion

### 3.1. Crystal Structure of 1-HTP and 1-LTP

To study the dynamic nature of the cyclic tetravanadate [(CH_3_)_3_CNH_3_]_4_[V_4_O_12_] by polymerizing into chains of metavanadate [VO_3_]_n_^n−^, we decided to study its behavior at low temperatures. To determine whether the title compound was able to undergo a single-crystal-to-single-crystal transformation at temperatures below 273 K, the modification of its crystal arrangement was followed by single-crystal X-ray diffraction measurements (sc-XRD) at all cooling or heating intervals (Table 1).

This tetravanadate-based compound has two different phases depending on the temperature at which the crystal structure measurement is carried out. Thus, at temperatures above 153 K, it crystallizes in the *I4/m* space group, whereas at lower temperatures (below 143 K), it crystallizes in the *I4_1_/a* space group. Therefore, from this point on, they will be named as **1-HTP** (High Temperature Phase) and **1-LTP** (Low Temperature Phase), respectively.

The asymmetric unit of [(CH_3_)_3_CNH_3_]_4_[V_4_O_12_] (**1-HTP**), already reported by us [23], consists of a quarter of a discrete cyclic anion [V_4_O_12_]^4-^ and a *tert*-butylammonium cation, [(CH_3_)_3_CNH_3_]^+^ (Figure 1a). The tetrameric anion on **1-HTP** is formed by four corner-linkers {VO_4_} linked by bridging oxygen atoms, arranged as a ring with its center located on a crystallographic 4-fold axis, and which displays a C4h symmetry. The vanadium and the bridging oxygen atoms form a square-like {V_4_O_4_} ring as represented in Figure 1. The terminal oxygen atoms that do not take part in the arrangement of the central {V_4_O_4_}, are placed at 1.287(2) Å above and below the ring plane. 

In **1-LTP**, the cell volume unit is 8 times the one for **1-HTP** and, thus, the asymmetric unit displays a complete cyclic anionic {V_4_O_12_}, with C_1_ symmetry, and four *tert*-butylammonium cations (Figure 1b). As shown in Figure 2b, the vanadium atoms together with two of the bridging oxygen atoms are still coplanar, whereas the two remaining ring oxygen atoms, O34 and O12, are located 0.23 and 0.47 Å away from the plane, respectively. This displacement of the ring implies changes in the hydrogen bonds between *tert*-butylammonium anions and the terminal oxygens, with the loss of a NH···O bond and the consequent rearrangement of the crystal packing. The distances and angles of the V-O bonds are shown in Appendix A in the Appendix A.

The crystal packing of **1-HTP** can be described as *cyclo*-tetravanadate anions stacking in (000) and (½ ½ ½) planes, which are further linked by the *tert*-butylammonium cations forming chains along the *z* axes. Each {V_4_O_12_} unit is connected to contiguous anions by strong hydrogen bonds between the nitrogen atoms of the cation and terminal oxygen atoms, as shown in Figure 2 and Figure 3. The crystal packing of **1-LTP** is similar to the one for **1-HTP**, however, the displacement of the oxygen atoms up and down in relation to the cycle plane undergoes through a rotation in a helical quaternary axis (Figure 3 and Appendix A). The displacement of O12 and O34 together with the rotation of the tetravanadate anion implies significant modifications in hydrogen bonds as well as in the unit cell. In this context, the *c* parameter shows a more dramatic change than *a* and *b* parameters, as the movement of the cycle oxygens and the hydrogen bond disappearance undergoes mainly among the *z* axis (Appendix A), even though the distance between centroids of the cycle does not experience a significant change from 7.353Å in **1-HTP** to 7.337Å in **1-LTP**.

Consequently, these H-bonds allow the structure to form chains through the V_4_O_12_ cycles, where the holes in each polyanionic chain means a 4.6% and a 4.8% of the total cell volume of **1-HTP** and in **1-LTP**, respectively (Figure 4).

### 3.2. Differential Scanning Calorimetry (DSC)

In order to understand the reversibility of the crystal transition, a Differential Scanning Calorimetry (DSC) of the compound [(CH_3_)_3_CNH_3_]_4_[V_4_O_12_] was also performed. As shown in the Figure 5, the Single-Crystal to Single-Crystal transition from a *I4_1_/a* space group to a *I4/m* space group showed a broad endothermic peak, by heating the sample and an exothermic peak in the opposite process of cooling. This process is a consequence of the rearrangement of the oxygen atoms and the loss hydrogen bonds described before.

This process occurs at 147 K on cooling the sample and at 157 K upon heating it. This measurement is in concordance with the crystal data discussed before. Additionally, the significant differences in the two Single-Crystal to Single–Crystal transformation temperature is due to the hysteresis of the sample. 

On the other hand, the enthalpy found for the Single-Crystal-to-Single-Crystal transformation of the cooling and the heating process are of equal value, this is 46 mJ/mg but showing an endothermic peak, in concordance with the reversibility of the transition. In this context, the SCSC transition on cooling is endothermic.

## 4. Conclusions

We have successfully characterized thermal behavior of a previously described polyoxovanadate compound ([(CH_3_)_3_CNH_3_]_4_[V_4_O_12_]), which undergoes a Single–Crystal–to–Single–Crystal structural transformation, the latter herein described. This compound, named [(CH_3_)_3_CNH_3_]_4_[V_4_O_12_], undergoes three consecutive and reversible phase transitions that can be followed by Single–Crystal X–Ray diffraction and take place on the range of 157 K on heating and 147 K on cooling. These transformations result in an endothermic transition from *I4_1_/a* space group to a *I4/m* one, and vice versa, which drastically change of the *c* cell parameter and create a reduction of the porosity.

## Figures and Tables

**Figure 1 materials-15-05663-f001:**
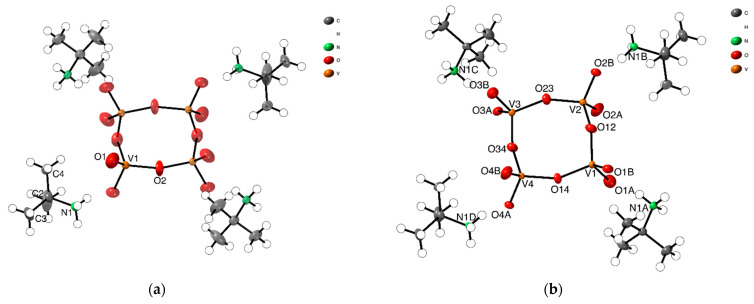
ORTEP representation of asymmetric units of (**a**) **1-HTP** at high temperatures, and (**b**) **1-LTP** at low temperatures, showing the 50% probability thermal ellipsoid.

**Figure 2 materials-15-05663-f002:**
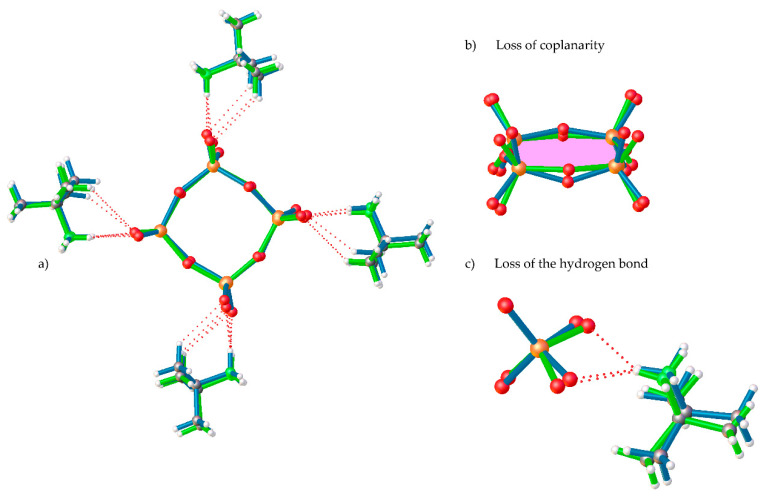
Overlay of (**a**) complete [(CH_3_)_3_CNH_3_]_4_[V_4_O_12_] compound, (**b**) detail of loss of coplanarity, and (**c**) detail of the loss of the NH···O of **1-HTP** (green) and **1-LTP** (blue).

**Figure 3 materials-15-05663-f003:**
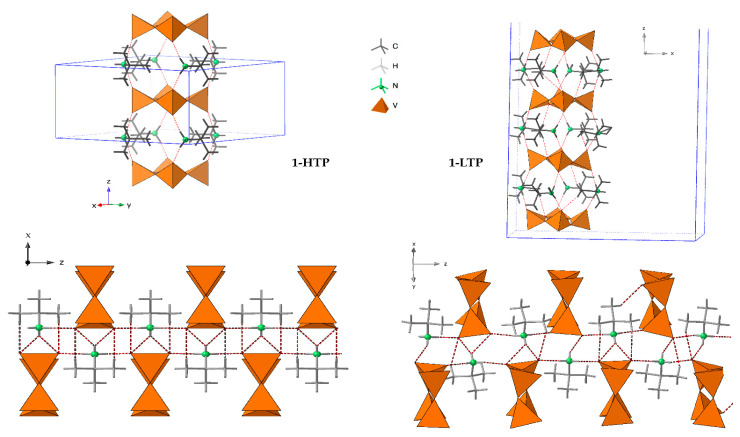
Crystal packing of **1-HTP** at the left and **1-LTP** at the right.

**Figure 4 materials-15-05663-f004:**
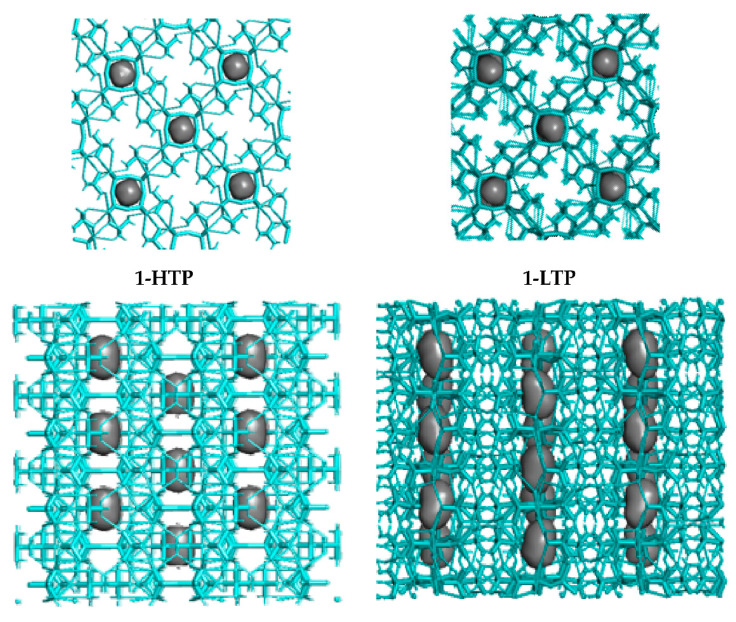
Crystal packing (top and side views of the holes) in compound **1-HTP** at the left and compound **1-LTP** at the right.

**Figure 5 materials-15-05663-f005:**
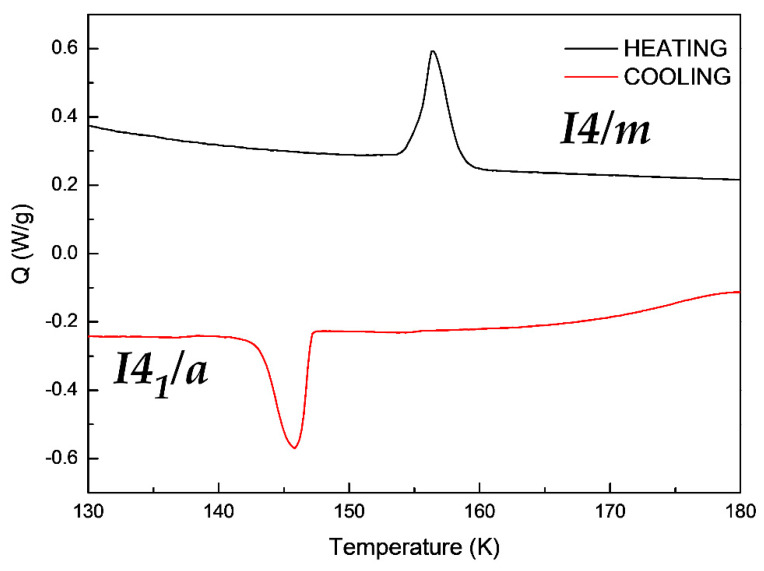
Differential Scanning Calorimetry measurements of compound [(CH_3_)_3_CNH_3_]_4_[V_4_O_12_].

**Table 1 materials-15-05663-t001:** Crystallographic data for [(CH_3_)_3_CNH_3_]_4_[V_4_O_12_] at different temperatures.

**Identification Code**	**1c_173**	**1c_163**	**1c_153**	**1c_143**	**1c_133**
empirical formula	C_16_H_48_N_4_O_12_V_4_	C_16_H_48_N_4_O_12_V_4_	C_16_H_48_N_4_O_12_V_4_	C_16_H_48_N_4_O_12_V_4_	C_16_H_48_N_4_O_12_V_4_
Fw (g mol^–1^)	692.34	692.34	692.34	692.34	692.34
temperature/K	173.14 (10)	163.13 (10)	153.13 (10)	143.14 (10)	133.15 (10)
space group	I4/m	I4/m	I4/m	I4_1_/a	I4_1_/a
*a* (Å)	14.97428(17)	14.96277(16)	14.9520(3)	21.0769(3)	21.1001(3)
*b* (Å)	14.97428(17)	14.96277(16)	14.9520(3)	21.0769(3)	21.1001(3)
*c* (Å)	7.35301(16)	7.34665(16)	7.3425(3)	29.3023(9)	29.3409(8)
V (Å^3^)	1648.76(4)	1644.80(4)	1641.51(8)	13,017.1(5)	13,063.0(5)
Z	2	2	2	16	16
*ρ*_calc_ (g cm^–3^)	1.395	1.398	1.401	1.413	1.408
*μ* (mm^–1^)	1.150	1.153	1.155	1.166	1.161
collected reflections	6123	6124	6147	47332	47559
unique reflections (Rint)	956 (0.034)	958 (0.032)	981 (0.037)	6756 (0.070)	6791 (0.070)
parameters	65	64	64	342	342
*R*(*F*) ^a^ [*I* > 2*σ*(*I*)]	0.034	0.033	0.047	0.051	0.048
*wR*(*F*^2^) ^b^ [all data]	0.081	0.083	0.120	0.161	0.147
Goodness-of-fit on F^2^	1.070	1.058	1.106	1.060	1.058

**Identification Code**	**1h_143**	**1h_153**	**1h_163**	**1h_173**	**1h_RT**
empirical formula	C_16_H_48_N_4_O_12_V_4_	C_16_H_48_N_4_O_12_V_4_	C_16_H_48_N_4_O_12_V_4_	C_16_H_48_N_4_O_12_V_4_	C_16_H_48_N_4_O_12_V_4_
Fw (g mol^–1^)	692.34	692.34	692.34	692.35	692.34
temperature/K	143.15 (10)	153.15 (10)	163.14 (10)	173.14 (10)	279 (30)
space group	I4_1_/a	I4_1_/a	I4/m	I4/m	I4/m
*a* (Å)	21.0904(3)	21.0988(3)	14.96239(17)	14.98187(17)	14.9856(2)
*b* (Å)	21.0904(3)	21.0988(3)	14.96239(17)	14.98187(17)	14.9856(2)
*c* (Å)	29.3206(9)	29.3229(9)	7.34681(16)	7.35678(16)	7.3653(2)
V (Å^3^)	13,042.0(5)	13,053.4(5)	1644.75(4)	1651.28(5)	1654.01(6)
Z	16	16	2	2	2
*ρ*_calc_ (g cm^–3^)	1.410	1.409	1.398	1.3923	1.390
*μ* (mm^–1^)	1.163	1.162	1.153	1.149	1.147
collected reflections	47334	47470	6131	6292	6087
unique reflections (Rint)	6769 (0.069)	6785 (0.070)	958 (0.032)	949 (0.033)	937 (0.036)
parameters	342	342	64	63	65
*R*(*F*) ^a^ [*I* > 2*σ*(*I*)]	0.051	0.054	0.034	0.033	0.033
*wR*(*F*^2^) ^b^ [all data]	0.160	0.173	0.084	0.079	0.089
Goodness-of-fit on F^2^	1.061	1.055	1.064	1.028	1.055
^a^*R*(*F*)=*Σ*||*F*_o_–*F*_c_||/*Σ*|*F*_o_|; ^b^*wR*(*F*_2_)={Σ[*w*(*F*_o_^2^–*F*_c_^2^)^2^]/*Σ*[*w*(*F*_o_^2^)^2^]}^1/2^					

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
