# Peer review of "Reversible First-Order Single Crystal to Single Crystal Thermal Phase Transition in [(CH3)3CNH3]4[V4O12]"

_materials, 2022, doi:10.3390/ma15165663_

Round 1

Reviewer 1 Report

The manuscript presents a detailed crystallographic study and data for reversible thermal phase transition of the compound tetrakis(tert-butylammonium)-cyclo-tetrametavanadate ([(CH3)3CNH3]4[V4O12]). XRD measurements were carried in heating (from 143 K to 183 K) and cooling regime (from 173 K - 133 K) and structural refinement on the obtained data frames was carried in order to obtain data for the nature of this reversible phase transition. The authors have identified that the material under study undergoes a reversible phase transition from the I41/a space group into the I4/m one around 157 K during heating, and 147 K during cooling. DSC measurements were carried to complement the crystallographic structure data and it was revealed that the phase transition is endothermic on cooling. 

Now, I believe that the manuscript has its merits. Even though I feel, personally, that it is a bit too specialised for Materials and probably it would have been a better fit in a more crystallography oriented journal. I also believe that, since polyoxometalates find so many applications (from antibacterials, to gas storage media, catalysts and semiconductors etc), and their structure is often not understood in depth, this manuscript could find its readership, if published in Materials. So, in general I can recommend it for consideration.

On the positive side - it is relatively well written, brief to the point, and even though it seems to lack abundance of references to others works, the provided crystallographic data would be of use to some researchers. However, I also feel that for such a short manuscript there are some confusing points, regarding the presentation.

I have listed a few points below. 

1) It is a inconsistent that the compound is referred to with a IUPAC name once, only in the abstract, re-defined as Compound (1) there (and in the supporting info) and then referred to either with the chemical formula (in two forms) or as a polyoxovanadate throughout the text. Especially, given that the topic of the paper is one single material.

2) The keywords could be improved.

3) The sample notation system is somehow confusing (codes listed in Table 1 in the form 1(h/c)-1_3). I am convinced that it is amazing as a placeholder for the filenames of the data the authors have obtained, but to guide the reader - they may feel a bit lost. I suggest some rearrangements in section 2.3. and that the sample designations are relocated at its end (just before Table 1). In its current state it predates the explanation about what high- and low- temperatures imply in the case of the measurements, obtained during temperature increase or decrease. Table 1 summarises results and probably below in the Results section anyway (where it is referred to once again).

4) FTIR spectra can be provided in the supporting materials. Now, there is only one mention of FTIR in a single sentence in the entire manuscript. 

5) It is unclear in what mode the XRD data was obtained, nor in what form were the samples - it is mentioned in the introduction that powder XRD was carried, the diffractometer mentioned in section 2.3. is a single-crystal one, however it be used for powder measurements. Please provide more details. And lines 62-70 should be improved for a better clarity and distinction about what was done in previous work [23] and in the current one. 

6) In the footnote of Table 1 add clarifying information about some of the parameter notations. Since Materials is not a crystallography-oriented journal, this may help the readers. 

7) ORTEP is not defined in the caption of Figure 1. Add a line in section 2.3. to clarify which software was used for the ORTEP-like presentation.

8) In the DSC analysis an enthalpy value of 46 mJ is shown, please clarify if it is mJ/g. Also correct the title of section 3.1. to “Differential … Calorimetry”

9) There are some minor typos in the text, which could be corrected with a more careful read-though: e.g. the wrongful use of the word “suffer” on multiple occasions, “opposite sing” in line 181 “147 _K” in line 191.

Author Response

1) It is an inconsistent that the compound is referred to with a IUPAC name once, only in the abstract, re-defined as Compound (1) there (and in the supporting info) and then referred to either with the chemical formula (in two forms) or as a polyoxovanadate throughout the text. Especially, given that the topic of the paper is one single material.

We agree with the reviewer that the way of naming the compound could have been confusing. Therefore, we changed the way of naming our compound along the text, referring to it only as “Compound 1h_RT”, and its derivatives (1-HTP and 1-LTP) or by its chemical formula “[(CH3)3CNH3]4[V4O12].

2) The keywords could be improved.

We agree with the reviewer that the keywords could be more related to the topic and thus, we replaced them. The new keywords have been added to the manuscript as follows:

“Polyoxovanadates, SCSC transformations, Thermal and structural characterization

3) The sample notation system is somehow confusing (codes listed in Table 1 in the form 1(h/c)-1_3). I am convinced that it is amazing as a placeholder for the filenames of the data the authors have obtained, but to guide the reader - they may feel a bit lost. I suggest some rearrangements in section 2.3. and that the sample designations are relocated at its end (just before Table 1). In its current state it predates the explanation about what high- and low- temperatures imply in the case of the measurements, obtained during temperature increase or decrease. Table 1 summarises results and probably below in the Results section anyway (where it is referred to once again).

Considering the comments made by Reviewer 1, we modified the “Section 2.3”:

Crystallographic data for the low and high temperature phases of the [(CH3)3CNH3]4[V4O12] compound are summarized in Table 1. For single-crystal X-ray diffraction measurements, the crystals were heated at different temperatures; i) cool-ing from 173 to 133 K every 10 K, and ii) heating from 143 to 183 K every 10 K, and then quenched to 100 K to 150 K for the data collection. Intensity data were collected on an Agilent Technologies SuperNova diffractometer, equipped with monochromatic Mo Kα radiation (λ = 0.71073 Å) and Eos CCD detector.

And the following text was added after Table 1:

The single-crystals were subjected to a cooling-heating procedure before X-ray data collection. In this way, the samples that were collected under the cooling process of the samples were named as; 1c_173, 1c_163; 1c_153; 1c_143; 1c_133, and the ones collected throughout the heating process were named as; 1h_143, 1h_153, 1h_163, 1h_173, 1h_RT. Nevertheless, for more clarity, we classify these 10 measurements into two groups; the crystallographic phase obtained at high temperatures (1-HTP), which exhibit the space group I4/m and comprises the identification codes 1c_173, 1c_163; 1c_153, 1h_163, 1h_173, 1h_RT; and the crystallographic phase obtained at low temperatures (1-LTP), which exhibit the space group I4/a and comprises the identification codes 1c_143; 1c_133, 1h_143, 1h_153.

4) FTIR spectra can be provided in the supporting materials. Now, there is only one mention of FTIR in a

single sentence in the entire manuscript.

We agree with the reviewer, this data has been added as supplementary information (as synthesis and characterization) as the FTIR is reported in a previous publication of the group, as well as cited (citation 23).

5) It is unclear in what mode the XRD data was obtained, nor in what form were the samples - it is mentioned in the introduction that powder XRD was carried, the diffractometer mentioned in section 2.3. is a single-crystal one, however it be used for powder measurements. Please provide more details. And lines 62-70 should be improved for a better clarity and distinction about what was done in previous work [23] and in the current one.

As it has been explained in lines 59-65, even if the crystallinity is lost when single-crystals of 1h_RT are heated, during the cooling process, the crystallinity is retained and therefore, single-crystal X-ray diffraction (sc-XRD) was used for the structural characterization of the different phases of [(CH3)3CNH3]4[V4O12]. Anyways, considering the comments made by Reviewer 1, we modify lines 59-70 in order to clarify that we follow the structural changes by sc-XRD. Thus, lines 59-70 were modified as follows:

Herein, ……..

6) In the footnote of Table 1 add clarifying information  about some of the parameter notations. Since Materials is not a crystallography-oriented journal, this may help the readers.

The data shown in table 1 is the common data shown when a structure is characterized by sc-XRD technique. The data consists on the empirical formula, molecular weight, cell parameters, and different parameters to ensure the good quality of the structure. If some of the parameters is not adequate it would be described and explained along the main text. As no comments are done, it is supposed that all the parameters agree in good agreement with a good resolution of the structure.

7) ORTEP is not defined in the caption of Figure 1. Add a line in section 2.3. to clarify which software was used for the ORTEP-like presentation.

The ORTEP is defined as a drawing for crystal structures which is representative of the thermal ellipsoids resulting from the data collected by sc-XRD. It is usually not necessary to add this definition, but considering the comments of Reviewer 1, the following lines were added to the Section 2.3. in order to briefly explain how the ORTEP representation was done:

The ORTEP representation of 1-HTP and 1-LTP were also represented in the CrystalMaker, showing a 50% probability thermal ellipsoids.

Additionally, we also modify the title of Figure 1 for more clarity:

Figure 1. ORTEP representation of asymmetric units of a) 1-HTP at high temperatures, and b) 1-LTP at low temperatures, showing the 50% probability thermal ellipsoid.

8) In the DSC analysis an enthalpy value of 46 mJ is shown, please clarify if it is mJ/g. Also correct the title of

section 3.1. to “Differential … Calorimetry”.

We agree with the reviewer, it  is mJ/mg. We have corrected both in the manuscripts.

9) There are some minor typos in the text, which could be corrected with a more careful read-though: e.g. the wrongful use of the word “suffer” on multiple occasions, “opposite sing” in line 181 “147 _K” in line 191.

We agree with the reviewer, thus, the following changes have been made:

Line 67  “Therefore, these materials, which suffers a structural change with” for “Therefore, these materials, which are able to undergo a structural change”.

Line 150 “In this context, the c parameter suffers a more dramatic change than a and b parameters” for “In this context, the c parameter shows a more dramatic change than a and b parameters”.

Line 152 “Even though, the distance between centroids of the cycle does not suffer a significant change from 7.353Å in 1-HTP to 7.337Å in 1-LTP” for “Even though, the distance between centroids of the cycle does not experience a significant change from 7.353Å in 1- HTP to 7.337Å in 1-LTP”.

Line 186-188 “We have successfully characterized thermal behavior of a previously described polyoxovanadate compound ([(CH3)3CNH3]4[V4O12]), these one suffers a Single–Crystal–to–Single–Crystal structural transformation” for “We have successfully characterized thermal behavior of a previously described polyoxovanadate compound ([(CH3)3CNH3]4[V4O12]), which undergoes a Single–Crystal–to–Single–Crystal structural transformation”.

Line 181 “this is 46 mJ/mg but with opposite sing, in concordance with the reversibility of the transition” for “this is 46 mJ/mg but showing an endothermic peak, in concordance with the reversibility of the transition”.

Line 191: we removed the underscore “147_K” for “147 K”.

Reviewer 2 Report

The paper «Reversible first-order single crystal to single crystal thermal 2 phase transition in [(CH3)3CNH3]4[V4O12]» describes the phase transition of a vanadate, and it deals with the characterization of the two phases following a temperature change, mainly form the point of view of physical properties. The paper is suitable for the publication in MDPI Materials, but the paper needs major revisions related to different aspects.

0. USE OF ENGLISH. English needs a revision, because some typos are present. Here you are a short list, some more checks are however required.

For example, in lines 38-40: «the crystal packing of the second phase, and, which allows to (38) observe the changes that undergo that kind of structural transitions ....» "and" in this case is not needed.

line 47: «of both, physical and chemical properties,» the comma after "both" is not needed.

line 58 and following: «we report» and other similar forms: it should be in the passive form, not in the personal form.

line 65: «into a catena form» do you mean a chain form? open chain, closed chain?

line 69 and following:«...reversible and involves the loss of coplanarity of the O-V-O bonds occurs through a thermal stimulus...» should be modified with «...reversible and involves the loss of coplanarity of the O-V-O bonds, WHICH occurs through a thermal stimulus...»

line 144: «strong hydrogen bonds, which implies nitrogen atoms of the cation and terminal oxygen atoms (Figure 2 and 3).» please, reformulate this sentence in a better way.

line 146: please, improve the English of the following sentence: «....however, the movement of the oxygen atoms up and down in relation to the cycle plane undergoes through....»

1. GENERAL COMMENTS. The paper is interesting, but it is poor from the point of view of the presented data. The authors present a deal of information about the crystallographic and structural data of the different phases involved in the transformation.
I do not understand if they are from the literature, or they are from the experimental results that the author obtained from the measurement; for this reason, FTIR spectra and diffractogram patterns for the different found phases should also be presented, to show the impact of the experimental work carried out in the framework of the paper.

2. PUNCTUAL COMMENTS line 73: «The compound [(CH3)3CNH3]4[V4O12] was synthesized according to literature methods» ok, but which was the shape and amount used for the characterization techniques of the present paper?

line 82: X–ray Crystallography was used as a characterization tool: power and voltage used, amount of material and which form, instrument configuration, steptime and stepsize should be provided as information, as well.

Author Response

  1. USE OF ENGLISH. English needs a revision, because some typos are present. Here you are a short list, some more checks are however required. For example;

- in lines 38-40: «the cc and, which allows to (38) observe the changes that undergo that kind of structural transitions ....» "and" in this case is not needed.

- line 47: «of both, physical and chemical properties,» the comma after "both" is not needed.

- line 58 and following: «we report» and other similar forms: it should be in the passive form, not in the personal form.

- line 65: «into a catena form» do you mean a chain form? open chain, closed chain?

- line 69 and following:«...reversible and involves the loss of coplanarity of the O-V-O bonds occurs through a thermal stimulus...» should be modified with «...reversible and involves the loss of coplanarity of the O-V-O bonds, WHICH occurs through a thermal stimulus...»

- line 144: «strong hydrogen bonds, which implies nitrogen atoms of the cation and terminal oxygen atoms (Figure 2 and 3).» please, reformulate this sentence in a better way.

- line 146: please, improve the English of the following sentence: «....however, the movement of the oxygen atoms up and down in relation to the cycle plane undergoes through....»

We agree with the reviewer that there are several mistakes in the use of english, we have corrected them. In some cases, the whole paragraph is highlighted, as it is included in other revisions (mistake line 58, 65, 69). In other cases (mistakes in line 38, and 47) have been erased. The mistake line 144 and 146 have been- re-written.

  1. GENERAL COMMENTS. The paper is interesting, but it is poor from the point of view of the presented data. The authors present a deal of information about the crystallographic and structural data of the different phases involved in the transformation. I do not understand if they are from the literature, or they are from the experimental results that the author obtained from the measurement; for this reason, FTIR spectra and diffractogram patterns for the different found phases should also be presented, to show the impact of the experimental work carried out in the framework of the paper.

The FTIR is reported in a previous publication of the group, as well as cited (citation 23). However, this data has been added as supplementary information (as synthesis and characterization) in order to avoid the confusion.

  1. PUNCTUAL COMMENTS line 73: «The compound [(CH ) CNH ] [V O ] was synthesized according to literature methods» ok, but which was the shape and amount used for the characterization techniques of the present paper?

line 82: X–ray Crystallography was used as a characterization tool: power and voltage used, amount of material and which form, instrument configuration, steptime and stepsize should be provided as information, as well.

On the one hand, the amount of the sample used in characterization techniques is provided in the experimental section, as materials and methods (" 8,3 mg"). On the other hand, the Crystal X-Ray diffraction, as it is not a powered sample, must be a unique cristal. As a consequence, we do not provide it. However, we had added a review request an image of the sample in the supplementary section.
Additionally, in the X–ray Crystallography  section the diffractometer configuration is provided:

"Crystallographic data for the low and high temperature phases of the [(CH3)3CNH3]4[V4O12] compound are summarized in Table 1. For single-crystal X-ray diffraction measurements, the crystals were heated at different temperatures; i) cooling from 173 to 133 K every 10 K, and ii) heating from 143 to 183 K every 10 K, and then quenched to 100 K to 150 K for the data collection. Intensity data were collected on an Agilent Technologies SuperNova diffractometer, equipped with monochromatic Mo Kα radiation ...."

Reviewer 3 Report

1. What is the point of this manuscript? It is not really clear to me why would a reader want to read this manuscript? The abstract needs to be rewritten to appeal to the general audience. Write 2-3 sentences as to why this study is important and what are its applications. 

2. " Therefore, these materials have attracted great attention for their wide application in various fields as photonic devices, optoelectronic technology, digital processing, sensors, etc. [13-15]." How?? Don't provide such vague statements. 

3. The title needs to be changed to reflect that this is a simulation study. 

4. "In order to understand the reversibility of the crystal transition, a Differential Scanning Calorimeter (DSC) of the compound [(CH3)3CNH3]4[V4O12] was also performed." Did the authors fabricate any sample? If so, they much provide image of the samples. 

5.  "147 _K on cooling. "  what does this underscore mean?

Author Response

  1. What is the point of this manuscript? It is not really clear to me why would a reader want to read this manuscript? The abstract needs to be rewritten to appeal to the general audience. Write 2-3 sentences as to why this study is important and what are its applications.

We had re-written the abstract.

  1. " Therefore, these materials have attracted great attention for their wide application in various fields as photonic devices, optoelectronic technology, digital processing, sensors, etc. [13-15]." How?? Don't provide

such vague statements.

We have been more concrete in this sentence.

  1. The title needs to be changed to reflect that this is a simulation study.

We have performed each single crystal X-ray diffraction measurements at low temperatures, as a consequence there are not simulation data. In this context, we consider that this change is not necessary.

  1. "In order to understand the reversibility of the crystal transition, a Differential Scanning Calorimeter (DSC) of the compound [(CH3)3CNH3]4[V4O12] was also performed." Did the authors fabricate any sample? If so, they much provide image of the samples.

As shown in Section 2.1. Materials and Methods, “The compound [(CH3)3CNH3]4[V4O12] was synthetized according to literature and identified by infrared (FT-IR) spectroscopy [23]”.

In addition, we have also added the images of the sample to the supplementary material.

[23] A. S. J. Wery, J. M. Gutierrez-Zorrilla, A. Luque, M. Ugalde, P. Roman, Phase Transitions in Metavanadates. Polymerization of Tetrakis(tert-Butylammonium) cyclo-Tetrametavanadate. Chem. Mater. 1996, 8, 408-413.”

  1. "147 _K on cooling. " what does this underscore mean?

This is a typing mistake which is already corrected in line 191.

Round 2

Reviewer 2 Report

The paper can be published in its present form.

Reviewer 3 Report

The authors made suitable revisions to the manuscript. I am happy to recommend the publication of the manuscript.